# Serum Level of Cytokeratin 18 (M65) as a Prognostic Marker of High Cardiovascular Disease Risk in Individuals with Non-Alcoholic Fatty Liver Disease

**DOI:** 10.3390/biom13071128

**Published:** 2023-07-14

**Authors:** Sabrina Pagano, Stephan J. L. Bakker, Catherine Juillard, Robin P. F. Dullaart, Nicolas Vuilleumier

**Affiliations:** 1Division of Laboratory Medicine, Diagnostics Department, Geneva University Hospitals, 1205 Geneva, Switzerland; nicolas.vuilleumier@hcuge.ch; 2Department of Medicine Specialties, Medical Faculty, Geneva University, 1211 Geneva, Switzerland; catherine.juillard@unige.ch; 3Division of Nephrology, Department of Internal Medicine, University Medical Center Groningen, University of Groningen, 9700 RB Groningen, The Netherlands; s.j.l.bakker@umcg.nl; 4Division of Endocrinology, Department of Internal Medicine, University Medical Center Groningen, University of Groningen, 9700 RB Groningen, The Netherlands; dull.fam@12move.nl

**Keywords:** Cytokeratin 18, non-alcoholic fatty liver disease (NAFLD), fatty liver index (FLI), Framingham risk score (FRS), Systematic COronary Risk Evaluation 2 (SCORE2)

## Abstract

Alterations in apoptosis, as reflected by circulating Cytokeratin 18 (CK18), are involved in the progression of non-alcoholic fatty liver disease (NAFLD) to non-alcoholic steatohepatitis and atherogenesis. We aimed to explore the discriminant accuracy of Cytokeratin 18 (CK18, including M65 and M30 forms) for an elevated fatty liver index (FLI) as a validated proxy of NAFLD, and cardiovascular disease (CVD) risk in the general population. Both serum CK18 forms were measured using a commercial immunoassay in randomly selected samples from 312 participants of the PREVEND general population cohort. FLI ≥ 60 was used to indicate NAFLD. Framingham Risk Score (FRS) and the SCORE2 were used to estimate the 10-year risk of CVD. The Receiver Operating Characteristic (ROC) curve, linear/logistic regression models, and Spearman’s correlations were used. Intricate associations were found between CK18, FLI, and CVD risk scores. While M30 was the only independent predictor of FLI ≥ 60, M65 best discriminated NAFLD individuals at very-high 10-year CVD risk according to SCORE2 (AUC: 0.71; *p* = 0.001). Values above the predefined manufacturer cutoff (400 U/L) were associated with an independent 5-fold increased risk (adjusted odds ratio: 5.44, *p* = 0.01), with a negative predictive value of 93%. Confirming that NAFLD is associated with an increased CVD risk, our results in a European general population-based cohort suggest that CK18 M65 may represent a candidate biomarker to identify NAFLD individuals at low CVD risk.

## 1. Introduction

Non-alcoholic fatty liver disease (NAFLD) [1] is defined as hepatic steatosis (triglycerides > 5.5% of liver volume) arising in the absence of significant alcohol intake without evidence of injury or fibrosis [1].

NAFLD is currently the most common cause of chronic liver disease globally, with a 20–30% prevalence in the adult population that grows in obesity or diabetic patients up to 70–90%, with elevated annual medical costs worldwide [2,3]. NAFLD encompasses a histological spectrum ranging from simple steatosis to non-alcoholic steatohepatitis (NASH), and can progress to fibrosis and cirrhosis with complications such as decompensation and hepatocellular carcinoma [1,2,4,5].

NALFD, as an underlying liver disorder, is estimated to become the leading indication for liver transplantation [3]. To meet population-based NAFLD screening requirements, several laboratory-based algorithms have been developed [6] to identify individuals at NAFLD risk in the general population, such as the fatty liver index (FLI) [7,8,9]. NAFLD is nowadays considered an independent risk factor for several extra-hepatic chronic diseases [9,10,11,12] such as cardiovascular disease (CVD), hypertension, type 2 diabetes (T2D), and chronic kidney disease. With CVD being the most common cause of death among NAFLD patients [13], accurate CVD risk stratification is also of major clinical importance, where biomarker-based approaches are often pragmatically appealing on top of providing further physiopathological insights.

Cytokeratin 18 (CK18) is a cytoskeletal protein and the main intermediate filament family member expressed in the liver [14] and other epithelial tissues [15]. The CK18 full-length form is released from necrotic cells, whereas a caspase-cleaved fragment is a product of the structural changes that occur during apoptosis [15,16]. Soluble total and fragments of CK18 can be detected in human serum with enzyme-linked immunoassay (ELISA) [17,18] (Figure 1). The M65 assay measures total CK18, i.e., full-length and caspase-cleaved fragments of CK18 generated during cell necrosis and apoptosis [17,18] (Figure 1, panel b), while the M30 assay detects a neoepitope created in the caspase-3 cleaved 30kDa fragment [17] during cell apoptosis (Figure 1, panel c).

Besides other circulating biomarkers, such as proinflammatory cytokines, iron and ferritin, and adipose tissue-derived hormones, total CK18 and its fragments have been proposed as promising biomarkers of liver cell death. CK18 levels have been found to be elevated in patients with NAFLD [19,20] and may be useful to differentiate between NAFLD and NASH [21]. Furthermore, CK18 is known to be expressed in atherosclerotic lesions, and its serum/plasma concentration is associated with coronary artery disease [22,23] on top of being increased in different diseases associated with an increased cardiovascular risk, such as chronic kidney disease [24], T2D [25], and other diseases related to increased endoplasmic reticulum stress and oxidative stress, typical features of cardiometabolic disorders [26,27,28].

A recent study by Quian and colleagues showed that CK18 (M65) was independently and positively associated with cardiometabolic disorders, even after adjustment for the presence of NAFLD and other cardiovascular risk factors [29]. Notably, limited data are available regarding the ability of CK18 to predict NAFLD in the general population and to predict CVD risk among NAFLD subjects. Taken together, these observations point to CK18 as an appealing biomarker candidate to capture both NAFLD and atherogenesis-related hazards.

Accordingly, the present study was aimed at evaluating (i) the discriminant accuracy of CK18 in identifying individuals suspected of having NAFLD (FLI ≥ 60) and (ii) within the subset of individuals with an FLI ≥ 60, the predictive ability of CK18 levels to predict high 10-year CVD risk according to Framingham Risk Score (FRS) and the Systematic COronary Risk Evaluation (SCORE)-2 algorithms. For this purpose, we used data from the Prevention of Renal and Vascular Endstage Disease (PREVEND) general-based population cohort study (https://umcgresearch.org/w/prevend, accessed on 9 July 2023).

## 2. Materials and Methods

### 2.1. The PREVEND General Population Cohort

The PREVEND (Prevention of Renal and Vascular Endstage Disease) cohort is a large population-based study including 8592 individuals aged 28–75 years from the city of Groningen (The Netherlands) [30,31]. From them, 6066 participants completed the third screening PREVEND study round (2004–2007), where active infectious hepatitis and alcohol consumption were excluded by a detailed questionnaire and for which an extensive clinical and biological characterization is available. Participants were instructed to remain fasting from 22:00 PM the day before visiting the PREVEND facility. Blood was obtained by venipuncture from an antecubital vein after 15–30 min rest. Blood samples were collected on melting ice and then allowed to clot. Serum was prepared by centrifugation at 1000× *g* for 10 min. Shortly thereafter, serum aliquots were stored at −80 °C. Samples were sent on dry ice to the laboratory of dr. Pagano. Total storage time was about 4 years. Samples were thawed only once on melting ice for measurement of CK18 and biochemical parameters required to calculate the FLI and the CVD risk scores. For the purpose of the current study, we randomly selected 312 individuals with available serum samples for measurement of CK18 and the biochemical parameters required for the present study.

### 2.2. NAFLD Definition in PREVEND

Suspected NAFLD was ascertained using the Fatty Liver Index (FLI) and defined as an FLI ≥ 60 as a validated proxy to detect NAFLD in the general population [7,8,9,31,32].

The FLI is calculated according to the following formula: FLI = (e^0.953 × loge (triglycerides) + 0.139 × BMI + 0.718 × loge (GGT) + 0.053 × waist circumference − 15.745)^/(1 + e^0.953 × loge (triglycerides) + 0.139 × BMI + 0.718 × loge(GGT) + 0.053 × waist circumference − 15.745^) × 100, where GGT is gamma-glutamyltransferase.

### 2.3. CVD Risk Prediction Assessment

Absolute risk for 10-year CVD was computed using the Framingham heart risk (FRS) [33] or the Systematic COronary Risk Evaluation 2 (SCORE2) [34] algorithms.

FRS calculation is based on gender, age, systolic blood pressure, treatment for hypertension, smoking, presence of diabetes, total cholesterol, and HDL cholesterol [33]. According to latest recommendations, absolute CVD risk percentage over 10 years was classified as low risk (<10%), intermediate risk (10–20%), and high risk (>20%) [35,36].

SCORE2 is a recently computed algorithm derived, calibrated, and validated to predict 10-year risk of first-onset CVD in European populations. It is calculated based on sex, age, smoking status, systolic blood pressure, total cholesterol, and HDL cholesterol [34]. The SCORE2 algorithm assigned three risk categories: (1) low-moderate risk, (2) high risk, and (3) very-high risk [34].

### 2.4. Cytokeratin 18 Assessment

The ELISA measurements of the Cytokeratin 18 concentrations were performed using the PEVIVA M65^®^ ELISA and M30^®^ ELISA kits (TECO medical AG, Sissach, Switzerland), according to their corresponding protocols. Absorbance was measured with the FilterMax F3 Multi-Mode Microplate Reader, using the SoftMax Pro software, version 7.0.3.

For the M65 ELISA test, a cutoff < 400 U/L was established on 222 normal subjects, with the 95th percentile equal to 413 U/L. M65 values > 400 U/L are a strong indication of liver disease [37]. The LLOD and LLOQ for this test were 25 U/L and 67 U/L, respectively.

For the M30 ELISA test, a cutoff < 200 U/L was established on 200 normal subjects, with the 95th percentile equal to 251 U/L. M30 values > 200 U/L are a strong indication of liver disease, as reported on the technical information sheet [37,38]. The LLOD and LLOQ for this test were 20 U/L and 40 U/L, respectively.

### 2.5. Biomarkers Determinations

Total cholesterol, HDL cholesterol, and triglycerides (TG) were measured in serum using routine procedures on a Roche Modular P chemistry analyzer (Roche 8000/H Cobas), and low-density lipoprotein (LDL) cholesterol was calculated using the Friedewald formula. Glucose, gamma-glutamyltransferase (GGT), alkaline phosphatase (ALP), alanine aminotransferase (ALT), and aspartate aminotransferase (AST) were quantified on a Roche Modular Platform. Glucose was measured in plasma by dry chemistry (Eastman Kodak, Amsterdam, The Netherlands).

### 2.6. Statistical Analysis

Continuous variables were expressed as median and interquartile range (IQR) and categorical variables in numbers with percentages. Normality of distribution was tested with the Shapiro–Wilk test. Comparisons between the two groups were performed using a non-parametric Mann–Whitney U test or Chi-square test. Correlations analyses were carried out using Spearman rank correlation test. In linear regression analysis, non-normally distributed data were transformed into natural logarithmic values.

C-statistics analyses were used to evaluate the discriminant accuracies of CK18 for FLI ≥ 60 and very-high 10-year CVD risk according to FRS and SCORE2, and reported as area under the curve (AUC). Univariate and adjusted logistic regression analyses were performed to examine the association between FLI (continuous value) or CK18 (continuous or categorical values) and a high 10-year risk for CVD (according to SCORE2 or FRS scoring) [33,34,35,36] in case of significant AUC only. High FRS or very-high SCORE2 categories (described in the methods section) versus moderate and low-risk groups together in the same category have been set as the binary outcome. We used the cutoff of 200 U/L and 400 U/L for M30 and M65, respectively [37,38], as specified in the above paragraph. Adjusted analyses for continuous or categorical variables were performed only in case of signification in univariate model. These analyses were carried out in three pre-specified PREVEND subgroups consisting of (i) the overall randomly selected individuals, (ii) those with FLI ≥ 60, and (iii) those with FLI < 60. Results are reported with 95% confidence intervals (95% CI). Due to the predefined study endpoints and the exploratory nature of this work, adjustment for multiple testing was not performed. Statistical analyses were performed with Tibco Statistica software (version 13.5.0.17, TIBCO Software Inc., Palo Alto, CA, USA) on the PREVEND cohort; statistical significance was set at *p* < 0.05. Receiver operating curve (ROC) analysis and the reported values of sensitivity, specificity, and positive and negative predictive values were performed using Analyse-it Software, Ltd. (Leeds, UK). Spearman’s correlation coefficients (bar graphs) and CK18 level distributions dichotomized according to the FLI, FRS, and SCORE2 were performed using GraphPad Prism 9.0.1 software (GraphPad Prism, Boston, MA, USA). In this case, non-parametric unpaired Mann–Whitney U tests were used for group comparisons.

## 3. Results

### 3.1. Characteristics of the Study Subjects

The demographic characteristics of the 312 randomly selected PREVEND participants are summarized in Table 1. In order to analyze the associations between CK18 (both forms M30 and M65) with CVD risk in individuals from the PREVEND general population with NAFLD, we dichotomized PREVEND participants according to FLI values < or ≥60 [7,32].

112 out of 312 (35.8%) participants had an FLI ≥ 60. Men were less likely to have an FLI ≥ 60 than women (*p* = 0.003). Participants with FLI ≥ 60 were older and had a higher body mass index (BMI), waist circumference, weight, diastolic and systolic blood pressure, TG, plasma glucose, ALT, AST, GGT, and a lower HDL-c than those with FLI < 60. Total-c and LDL-c did not differ significantly between the two groups. Both CK18 M30 and M65 levels were significantly higher in subjects with FLI ≥ 60 than in subjects with FLI < 60. Moreover, the FLI ≥ 60 group had significantly higher values of FRS and SCORE2 compared to the FLI < 60 group (Table 1).

### 3.2. CK18 and the Risk of NAFLD (FLI ≥ 60)

On the 312 studied PREVEND participants, C-statistics analyses showed that both M30 and M65 as continuous variables had a significant discriminant accuracy in predicting an FLI ≥ 60, with AUCs of 0.702 (CI: 0.641–0.762; *p* < 0.0001) and 0.657 (CI: 0.59–0.719; *p* < 0.0001), respectively (Appendix A). Adjusted logistic regression analyses indicated that only M30 was an independent predictor of FLI ≥ 60. Values above the predefined cutoff of 200 U/L (provided by the manufacturer) were associated with an independent 3-fold increased risk of NAFLD (FLI ≥ 60) (Appendix A). At this cutoff, the sensitivity (SE), specificity (SP), and positive and negative predictive values (PPV, NPV) were 57.1%, 72.5%, 54.0%, and 75.0%, respectively.

### 3.3. CK18 and High-Risk of Cardiovascular Disease Prediction

To further explore the predictive strength of CK18 for CVD risk in PREVEND participants we performed C-statistics and logistic regression in the three pre-specified PREVEND groups (overall, FLI ≥ 60 and FLI < 60).

As shown in Table 2, FLI displayed significant AUCs to discriminate an individual at a high 10-year CVD risk according to FRS across the three considered groups. M65 was also a significant predictor of high CVD risk overall and in the FLI ≥ 60 groups, while M30 was found to bear some significant predictive ability in the overall cohort only (Table 2). Extending these observations, multivariate logistic regression analyses (using CK18 continuous or categorical values) indicated FLI was the only independent predictor of high FRS (*p* < 0.0001) in the overall cohort and the two subgroups. A close to significant association was observed for M65 in the overall and FLI ≥ 60 subgroups as a continuous variable, but not when categorized according to the pre-specified cutoff (Table 2). No associations were retrieved for M30 in any of the subgroups analyzed (Table 2).

As the SCORE2 algorithm was likely better calibrated for European populations than the FRS [39], we repeated the same analyses with the 10-year CVD risk computed according to SCORE2 as described in Table 3. In the overall group, C-statistics indicated that FLI as a continuous value was found to be the only significant predictor of a very high CVD risk, while a non-significant trend was noted for M65 (Table 3). On the other hand, in the FLI ≥ 60 subgroup, M65 was the only predictor of high CV risk according to SCORE2, with an AUC of 0.71. Logistic regression analyses corroborated these results by demonstrating that M65 was independently associated with a 5-fold increased risk of a very high CVD risk, according to SCORE2 (Table 3). M65 values above the pre-specified manufacturer cutoff were associated with the following SE, SP, PPV, and NPV: 41.7%, 89.7%, 33%, and 93.0%, respectively. None of these predictors of interest were found to be discriminant in the FLI < 60 subgroup (Table 3).

The distribution of CK18 M65 and M30 levels in overall PREVEND subjects dichotomized according to the FLI ≥ and <60, the FRS (FRS very high risk vs. low-moderate high risk) and SCORE 2 (SCORE2 very high-risk vs. low-moderate high risk) is illustrated in Figure 2. CK18 (M30) and CK18 (M65) levels were higher in participants with an FLI ≥ 60 vs. an FLI < 60 and in participants with a very high risk vs. participants with a low-moderate high risk according to FRS, but were not significantly different upon dichotomization according to SCORE2.

### 3.4. Determinants of CK18 Associations with Cardiovascular Disease Risk Scores and FLI

In the overall cohort, Spearman analyses indicated positive correlations between M30 and M65 with most of the hepatic and cardiometabolic parameters, including the FLI, FRS, and SCORE2 with inverse correlations for total-c, LDL-c, and HDL-c (Figure 3 panels a and b). Many of these associations were lost in the FLI ≥ 60 or FLI < 60 subgroups (Figure 3, panels c, d, e, and f). In the FLI ≥ 60 subgroup, the significant positive correlations between M30 and M60 with liver function enzymes were maintained, as well as the significant negative associations with total cholesterol and LDL-C. In the FLI < 60 subgroup, M30 and M65 were associated with plasma glucose, AST, and GGT. In addition, M30 was associated with systolic blood pressure (SBP), waist circumference, and the FLI (Figure 3, panel f).

In order to further delineate associations between M30 and M65 with FRS, SCORE2, or FLI, we performed univariate and multivariate linear regression analyses on the three groups of PREVEND participants (Table 4). In the 312 subjects, combined univariate linear regression analyses demonstrated that M30, M65, and FLI were associated with FRS or SCORE2 and that M30 and M65 were associated with FLI (Table 4). However, in multivariate analyses, only FLI was significantly associated with FRS, while both FLI and M65 were significantly associated with SCORE2 (Table 4). In addition, M30 and M65 were associated with FLI (β = 0.41, *p* = 0.0001; β = 0.23, *p* = 0.03, respectively).

In the FLI ≥ 60 group (Table 4), univariate linear regression analysis indicated that only the FLI was associated with the FRS while M65 and the FLI were associated with SCORE2; the associations of M30 and M65 with FLI were lost (β = 0.004, *p* = 0.88; β = 0.03, and *p* = 0.28, respectively). In multivariate analyses, the FLI was independently associated with the FRS, and the FLI and M65 were independently associated with SCORE2 (Table 4). The same analyses performed on participants with FLI < 60 highlighted that only the FLI was independently associated with FRS or SCORE2 and that only M30 was associated with the FLI (Table 4).

## 4. Discussion

The first important finding of the present study is that the FLI, a validated biochemically derived index recommended for large-scale NAFLD screening in the general population [7,9,32,40], represents an independent CVD risk predictor, further lending weight to the cumulative body of evidence showing that NAFLD is associated with an increased CVD risk [11,41,42,43]. Our current results showing an association between an elevated FLI and CVD risk according to FRS corroborate and extend recent findings derived from Korean [44] and European populations [45]. Our results also showed that the prevalence of elevated FLI in the general population is similar to what has been previously reported [9,32]. Interestingly, we found that in the context of an FLI ≥60, this independent association could not be reproduced with SCORE2, possibly due to model calibration differences between FRS and SCORE2 [39]. The FRS has been developed in the United States [33], while SCORE2 has been recently derived to estimate 10-year fatal and non-fatal cardiovascular disease (CVD) risk in individuals from Europe [34]. Whether a calibration or any other non-mutually exclusive issues underlie such a discrepancy in our European general population study is still unknown and warrants further scrutiny. The discordance in those results could be seen as a relevant finding of our study given that the PREVEND cohort originates from the North of Europe, and consequently, SCORE2 may be a more suitable algorithm to use for CVD risk estimation.

The second important finding of this work is that total CK18 (M65) was found to be associated with a very high CVD risk according to the SCORE2, both particularly in individuals with an FLI ≥ 60. Consistent with previous studies [25,29,46], our linear regression and Spearman analyses showed that M30 and M65 were independently associated with the FLI and with several cardiometabolic parameters, especially in the overall study population where M30 and M65 were significantly associated with almost all the parameters studied and with both CVD risk scores, while when considering the same associations in participants with FLI ≥ 60, M30 and M65 were significantly associated only with liver function enzymes (ALT, AST, and GGT) and only M65 was associated with FRS.

Extending previous observations reporting higher levels of CK18 (M30 and M65) in NAFLD subjects compared to healthy subjects [25,29,46] and even more in steatohepatitis [41], this is, to our knowledge, the first demonstration indicating that CK18 (M30) predicts FLI-suspected NAFLD, except for a very recently shown relationship of CK (M30) with the FLI as a continuous parameter [47], and that CK-18 (M65) predicts 10-year CVD risk independently of FLI if the SCORE2 algorithm is used for CVD risk stratification purposes.

Using the M65 pre-specified and previously validated cutoff set at 400 U/L, the odds of very high CVD would increase by 5-fold (adjusted OR: 5.44; *p* = 0.01). At this cutoff, the NPV of 93.0% indicates that M65 may potentially be useful in a primary care setting to exclude very-high-risk CVD in NAFLD individuals. The potential clinical application needs to be determined in larger multi-center studies.

NAFLD and CVD are both outcomes of end-organ damage caused by metabolic abnormalities commonly captured by metabolic syndrome components such as central obesity, high blood pressure, elevated glucose, elevated triglycerides, and low HDL-c. It is challenging to determine the exact impact of NAFLD per se on increased CVD risk, as both conditions share common risk factors that contribute to this heightened risk [48]. The connections between NAFLD and CVD involve intricate and interrelated mechanisms that operate through multiple pathways simultaneously [48].

In this scenario, CK18 (M30 and M65) could be a marker not only of liver cell death but also of damaged cardiomyocytes that lose cellular integrity during response to abnormal cellular stresses such as endoplasmic reticulum stress and oxidative stress, which are recognized as key features of cardiometabolic disorders [26,27].

Endothelial apoptosis can lead to endothelial dysfunction and the development of hypertension [49]. Apoptosis of cardiomyocytes is associated with both the aging process and chronic cardiac overload [50] and plays a substantial role in altering cardiac geometry and progressively deteriorating myocardial function, potentially leading to chronic cardiomyopathy and advanced heart failure [51]. A recent study reported that CK18 (M30) level was highly correlated to left ventricular (LV) diastolic dysfunction in adolescents with obesity [52], although CK18 (M65) had not been investigated. In addition, another study determined the relationship between the development of LV remodeling and CK18 (M30) but not CK18 (M65) in patients with anterior ST-segment elevation myocardial infarction [53]. In this study, the cutoff used for CK18 (M30) was 144.9 U/L, different from what we used (200 U/L), and they retrieved an AUC of 0.893 for CK18 (M30) level for predicting LV remodeling [53]. In an effort to assess patients with acute coronary syndrome (ACS), including unstable angina and acute myocardial infarction (AMI), as well as patients with stable angina, serum levels of CK18 (M30 and M65) were measured [23]. In addition, among patients with an acute coronary syndrome and stable angina, it was found that only CK (M30) levels accurately reflected the severity of coronary artery disease in acute myocardial infarction patients [23]. Collectively, these studies consistently indicate that CK (M30) is a more reliable marker for acute myocardial events.

Our findings emphasize the association between CK18 (M65) and a higher risk of CVD, likely due to the ability of CK (M65) to detect cell death across various forms, including apoptosis, necrosis, and autophagy, and not only apoptosis, as CK18 (M30) can detect.

In conclusion, the present results show that CK18 (M30) is a predictor of the FLI suspected NAFLD, confirming that the FLI is an independent predictor of a high 10-year CV risk according to both the FRS and the SCORE2 algorithm. Furthermore, this hypothesis-generating study indicates that CK18 (M65) measurement could help to exclude a very-high CVD risk in NAFLD individuals, based on a negative predictive value of 93% for CK18 (M65) > 400 U/L. Further research is needed to validate these findings, support M65’s clinical significance, and clarify these observed associations in a longitudinal design.

**Limitations of the study**. Firstly, our cross-sectional observational study included a relatively limited number of randomly selected PREVEND participants, raising the possibility of selection bias. Given the fact that both the prevalence of the FLI-based NAFLD suspicion in the present study and the FLI associations with cardiometabolic features were similar to what was previously reported using data from PREVEND and from other European populations [6,10,40], we consider that such an issue is unlikely to have blunted the present results. Nonetheless, our current preliminary findings should be considered to be hypothesis-generating. Secondly, although the FLI is an accepted diagnostic proxy for NAFLD categorization in population studies [5,6,7], it remains a surrogate indicator of hepatic fat accumulation, with liver biopsy being the most accurate and reliable tool for assessing the presence and severity of NAFLD. Thirdly, we could not further validate our findings by taking into account other well-validated indirect indices of liver fibrosis, such as the FIB-4 score [4], due to the lack of appropriate sampling for platelet counts upon study inclusion. For this reason, our study does not allow for a conclusion regarding the association of CK18 with liver fibrosis.

## Figures and Tables

**Figure 1 biomolecules-13-01128-f001:**
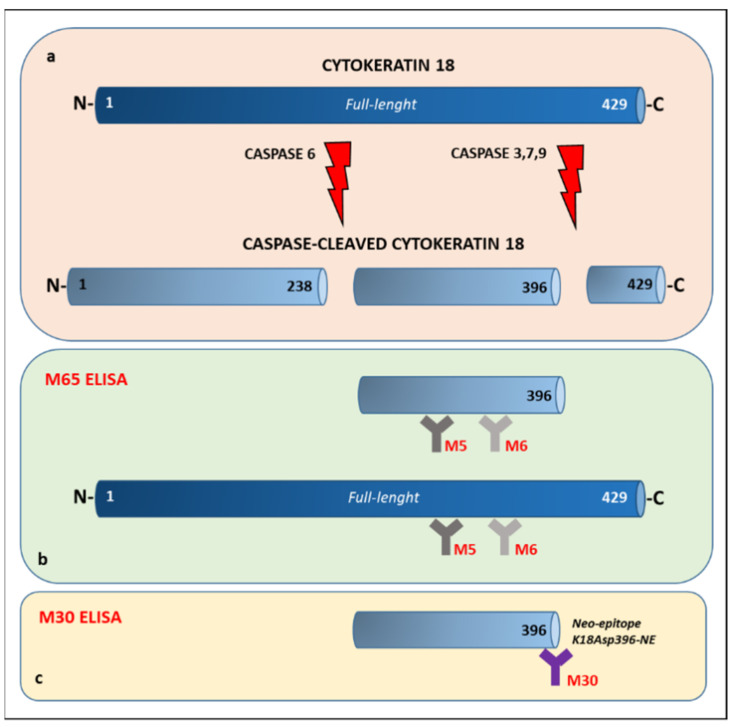
CK18 detection and distinction between M30 and M65 immunoassays. (**a**) Full-length and caspase-cleaved CK18 fragments. (**b**) The cytokeratin 18 (CK18) M65 immunoassay measures total CK18 that is, full-length and caspase-cleaved CK 18 fragments generated during cell necrosis and apoptosis. (**c**) CK18 M30 assay detects a neoepitope created in the caspase-3 cleaved 30-kDa fragment released during cell apoptosis only.

**Figure 2 biomolecules-13-01128-f002:**
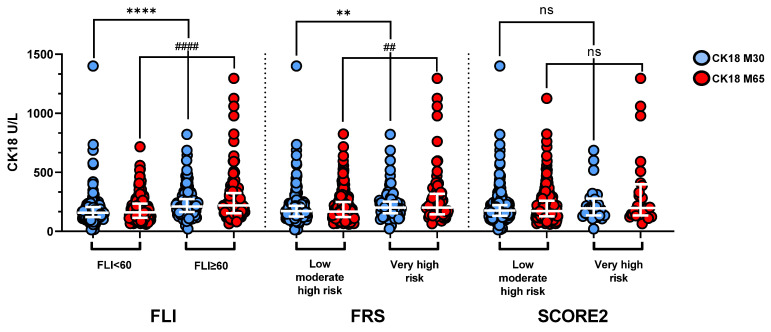
CK18 (M30 and M65) levels in PREVEND subjects dichotomized according to FLI, FRS, and SCORE2. M30 and M65 levels were higher in FLI ≥ 60 and FRS ≥ 20 groups compared to FLI < 60 and FRS < 20 groups but not different according to SCORE2. Bars indicate median and interquartile range. Results are expressed as individual values and bars. *p*-value by Mann–Whitney test. **** *p* < 0.0001 and #### *p* < 0.0001; ** *p* = 0.0027 and ## *p* = 0.0025; and ns: *p* = 0.14 (M30) and ns: *p* = 0.45 (M65).

**Figure 3 biomolecules-13-01128-f003:**
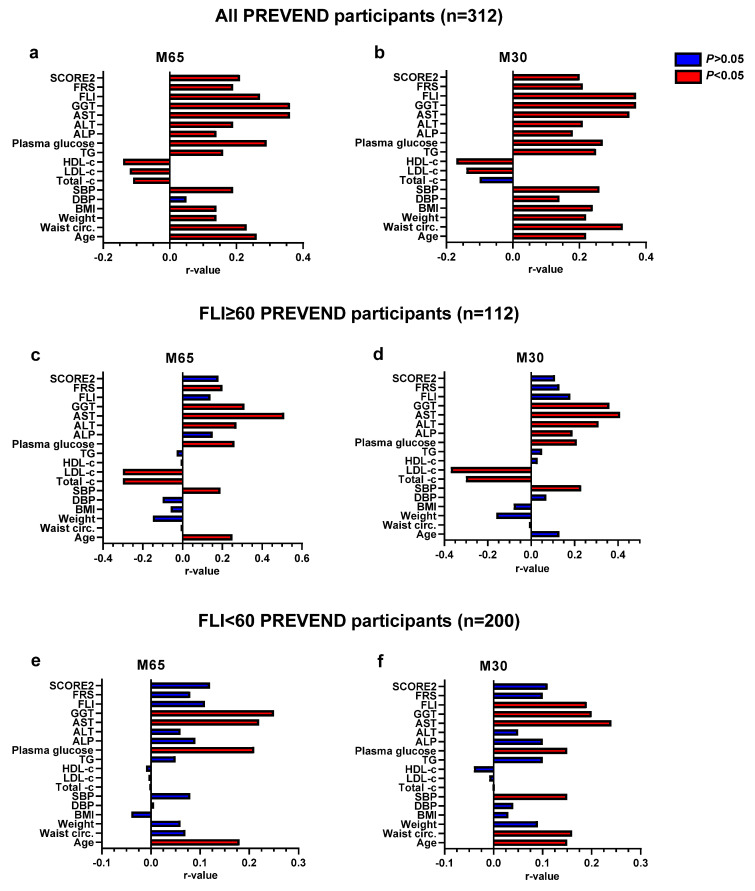
Spearman’s rank correlation between levels of CK18 (M65 and M30) and clinical characteristics of PREVEND participants. Panels (**a**,**c**,**e**) exhibit the correlations between M65 levels and clinical characteristics of all PREVEND participants, FLI ≥ 60 individuals, and FLI < 60 individuals, respectively. Panels (**b**,**d**,**f**) exhibit the correlations between M30 levels and clinical characteristics of all PREVEND participants, FLI ≥ 60 individuals, and FLI < 60 individuals, respectively. The bar charts show the r-values (positively and negatively) of CK18 (M65) and CK18 (M30). In red: significant correlation coefficients at *p* < 0.05. In blue: non-significant correlation coefficients (*p* > 0.05). Exact *p*-values are shown in Appendix A.

**Table 1 biomolecules-13-01128-t001:** Clinical characteristics of PREVEND participants (n = 312) according to FLI status.

	Overall (n = 312)	FLI < 60 (n = 200)	FLI ≥ 60 (n = 112)	*p*-Value
**Demographic**				
Age, yr.	53 (46–65)	51 (45–59.5)	59 (50.5–69)	<0.0001
Males, no. (%)	154 (49.3)	111 (55.5)	43 (38.3)	0.01
Waist circumference, cm	94 (85–104)	87 (80–94)	107 (102–114)	<0.0001
Weight, kg	78 (69.7–89)	73 (65–80)	93 (84–102)	<0.0001
BMI, kg/m^2^	26.5 (23.7–29.5)	24.6 (22.8–26.4)	30.4 (28.8–32.9)	<0.0001
Diastolic blood pressure, mm Hg	72 (67–78)	70 (65–76)	76 (71–80)	<0.0001
Systolic blood pressure, mm Hg	124 (111–138)	118 (108–130)	134 (123–146)	<0.0001
Current smoker, no. (%)	87 (27.9)	65 (32.5)	22 (19.6)	0.064
Type 2 diabetes, no. (%)	43 (13.7)	17 (8.5)	16 (14.2)	0.03
FRS (%) §	11.7 (6.1–23)	9.1 (5.3–14.9)	18.5 (11.7–30)	<0.0001
SCORE2 (%) §§	4 (2.1–7.2)	3.2 (1.9–5.8)	5.75 (3.5–9)	<0.0001
**Biochemical**				
Total-c mg/dL	195 (167–221.9)	196.8 (166.2–220.4)	189.8 (167.6–224.4)	0.97
LDL-c mg/dL #	131.7 (104–155.7)	132.3 (106–154.2)	123.7 (100.2–155.8)	0.46
HDL-c mg/dL	38.2 (31.7–47.5)	41.9 (35.5–50.8)	32.4 (26.9–37.1)	<0.0001
TG mg/dL	114.2 (84.1–157.6)	95.6 (69–123.1)	162 (120.4–215.2)	<0.0001
Plasma glucose, mg/dL	85 (77–92)	81.4 (76–88.6)	90 (84.6–106.2)	<0.0001
ALP, U/L	44 (35–54)	42 (34–52.5)	47.5 (40–56.5)	0.0008
ALT, U/L	6.5 (5–9)	5.8 (5–8)	7.9 (5.6–10.1)	<0.0001
AST, U/L	18 (15–23)	18 (14–21)	20 (16–26)	0.0005
GGT, U/L	21 (14–35)	16.5 (12.5–24.5)	36 (24–54.5)	<0.0001
FLI (%) §§§	41.3 (16.7–75.6)	22.2 (10.9–39.5)	82.3 (73.4–92.5)	<0.0001
CK-18/M30, U/L	176.9 (132.8–224.5)	158.6 (121.2–204.6)	210.6 (163.7–272.4)	<0.0001
CK-18/M65, U/L	173.1 (128.2–263.1)	161.1 (115.5–234.6)	220.4 (153.6–323.2)	<0.0001

All continuous variables are expressed as median (interquartile range [IQR]) or number [no.] (percentages [%]). *p*-value (Mann–Whitney U-test for continuous variables and Fisher’s exact test for categorical variables). Abbreviations: total-c: total cholesterol; LDL-c: low-density lipoprotein cholesterol; HDL-c: high-density lipoprotein cholesterol; TG: triglyceride; BMI: body mass index; ALT: alanine transaminase; AST: aspartate transaminase; FLI: fatty liver index; ALP: alkaline phosphatase; GGT: gamma-glutamyl-transferase; FRS: Framingham risk score; SCORE: Systematic COronary Risk Evaluation; and CK: cytokeratin. § FRS: calculated based on sex, age, smoking status, presence of diabetes, hypertension treatment, total cholesterol, and HDL cholesterol. §§ SCORE2: calculated based on sex, age, smoking status, systolic blood pressure, total cholesterol, and HDL cholesterol. §§§ FLI: (e ^0.953 × loge (triglycerides) + 0.139 × BMI + 0.718 x loge (ggt) + 0.053 × waist circumference − 15.745)^/(1 + e ^0.953 × loge (triglycerides) + 0.139 × BMI + 0.718 × loge (ggt) + 0.053 × waist circumference − 15.745^) × 100. # LDL-c: calculated according to Friedwald formula.

**Table 2 biomolecules-13-01128-t002:** Performance of FLI or CK18 (M30 and M65), categorical or continuous value in predicting high FRS in PREVEND participants.

	**All PREVEND Participants (n = 312)**								
	**Discriminant Accuracy**			**Univariate Analysis**			**Multivariate Analysis**		
**Predictor**	**AUC**	**95% CI**	***p-*Value**	**OR**	**95% CI**	***p-*Value**	**OR**	**95% CI**	***p-*Value**
M30	0.609	0.540–0.677	0.03	1.001	0.99–1.00	0.08	0.99	0.99–1.001	0.32
FLI	0.722	0.658–0.785	<0.0001	1.03	1.01–1.03	<0.0001	1.025	1.01–1.035	<0.0001
M65	0.608	0.540–0.677	0.03	1.003	1.001–1.004	0.001	1.002	0.99–1.004	0.05
M30 > 200	-	-	-	2.01	1.22–3.32	0.005	1.10	0.61–2.00	0.73
FLI	0.722	0.658–0.785	<0.0001	1.03	1.01–1.03	<0.0001	1.026	1.016–1.035	<0.0001
M65 > 400	-	-	-	2.01	0.91–4.45	0.08	1.10	0.57–3.51	0.44
	**FLI ≥ 60 PREVEND Participants (n = 112)**								
	**Discriminant Accuracy**			**Univariate Analysis**			**Multivariate Analysis**		
**Predictor**	**AUC**	**95% CI**	***p-*Value**	**OR**	**95% CI**	***p-*Value**	**OR**	**95% CI**	***p*-Value**
M30	0.572	0.466–0.679	0.187	-	-	-	-	-	-
FLI	0.678	0.580–0.777	0.001	1.06	1.023–1.010	0.001	1.06	1.020–1.100	0.002
M65	0.615	0.511–0.719	0.036	1.002	1.000–1.004	0.03	1.001	0.999–1.004	0.06
M30 > 200	-	-	-	0.52	0.246–1.135	0.102	-	-	-
FLI	0.678	0.580–0.777	0.001	1.06	1.023–1.010	0.001	1.062	1.023–1.102	0.001
M65 > 400	-	-	-	1.58	0.545–0.46	0.39	1.46	0.48–4.47	0.49
	**FLI < 60 PREVEND Participants (n = 200)**								
	**Discriminant Accuracy**			**Univariate Analysis**			**Multivariate Analysis**		
**Predictor**	**AUC**	**95% CI**	***p-*Value**	**OR**	**95% CI**	***p-*Value**	**OR**	**95% CI**	***p*-Value**
M30	0.543	0.447–0.639	0.416	-	-	-	-	-	-
FLI	0.618	0.520–0.716	0.025	1.024	1.003–1.045	0.02	-	-	-
M65	0.532	0.434–0.630	0.546	-	-	-	-	-	-
M30 > 200	-	-	-	1.109	0.506–2.43	0.79	-	-	-
FLI	0.618	0.520–0.716	0.025	1.024	1.003–1.045	0.02	-	-	-
M65 > 400	-	-	-	1.50	0.388–0.587	0.55	-	-	-

AUC: area under the curve; OR: odds ratio; CI: confidence interval; CK: cytokeratin; and FLI: fatty liver index. (-): where AUC is non-significant, logistic regression analysis was not performed, and where univariate logistic regression is non-significant, the variable is consequently not added in multivariate analysis. FLI is used as continuous variable. M30 > 200 and M65 > 400 means that variables are used as categorical predictors with the cutoff as specified in the methods section. In multivariate analysis, only FLI was included in the model, considering M30 and M65 as continuous or categorical.

**Table 3 biomolecules-13-01128-t003:** Performance of FLI or CK-18 (M30 and M65), categorical or continuous value) in predicting very-high CV risk according to SCORE2 in PREVEND participants.

	**All PREVEND Participants (n = 312)**								
	**Discriminant Accuracy**			**Univariate Analysis**			**Multivariate Analysis**		
**Predictor**	**AUC**	**95% CI**	***p-*Value**	**OR**	**95% CI**	***p-*Value**	**OR**	**95% CI**	***p-*Value**
M30	0.554	0.422–0.686	0.39	-	-	-	-	-	-
FLI	0.637	0.518–0.756	0.029	1.015	1.001–1.029	0.035	1.009	0.99–1.024	0.21
M65	0.608	0.478–0.737	0.085	1.003	1.001–1.004	0.0006	1.002	1.00–1.004	0.005
M30 > 200	-	-	-	1.267	0.537–2.991	0.587	-	-	-
FLI	0.637	0.518–0.756	0.029	1.015	1.001–1.029	0.035	1.012	0.991.027	0.08
M65 > 400	-	-	-	4.23	1.516–11.83	0.005	3.59	1.25–10.26	0.01
	**FLI ≥ 60 PREVEND Participants (n = 112)**								
	**Discriminant Accuracy**			**Univariate Analysis**			**Multivariate Analysis**		
**Predictor**	**AUC**	**95% CI**	***p-*Value**	**OR**	**95% CI**	***p-*Value**	**OR**	**95% CI**	***p*-Value**
M30	0.587	0.384–0.790	0.384	-	-	-	-	-	-
FLI	0.636	0.467–0.805	0.086	1.045	0.984–1.109	0.147	1.043	0.975–1.115	0.21
M65	0.714	0.524–0.904	0.016	1.003	1.001–1.005	0.001	1.003	1.001–1.005	0.002
M30 > 200	-	-	-	1.592	0.449–5.644	0.471	-	-	-
FLI	0.636	0.467–0.805	0.086	1.045	0.984–1.109	0.147	1.042	0.98–1.108	0.18
M65 > 400	-	-	-	5.584	1.50–20.65	0.009	5.444	1.44–20.53	0.01
	**FLI < 60 PREVEND Participants (n = 200)**								
	**Discriminant Accuracy**			**Univariate Analysis**			**Multivariate Analysis**		
**Predictor**	**AUC**	**95% CI**	***p-*Value**	**OR**	**95% CI**	***p-*Value**	**OR**	**95% CI**	***p*-Value**
M30	0.495	0.346–0.644	0.955	-	-	-	-	-	-
FLI	0.568	0.424–0.712	0.448	-	-	-	-	-	-
M65	0.500	0.376–0.625	0.996	-	-	-	-	-	-
M30 > 200	-	-	-	0.555	0.116–2.655	0.461	-	-	-
M65 > 400	-	-	-	1.618	0.189–13.80	0.659	-	-	-

SCORE: Systematic COronary Risk Evaluation; AUC: area under the curve; OR: odds ratio; CI: confidence interval; CK: cytokeratin; and FLI: fatty liver index. (-): where AUC is non-significant, logistic regression analysis was not performed, and where univariate logistic regression is non-significant, the variable is consequently not added in multivariate analysis. FLI is used as continuous variable. M30 > 200 and M65 > 400 means that variables are used as categorical predictors with the cutoff, as specified in the methods section. In multivariate analysis, only FLI was included in the model considering M65, M30 being non-significant in the univariate analysis.

**Table 4 biomolecules-13-01128-t004:** Linear regression analysis using CK18 (M30 and M65) and FLI as independent variables for FRS or SCORE2 as dependent variables and using M30 and M65 as independent variables for FLI as dependent variables in PREVEND participants.

	**All PREVEND Participants (n = 312)**			
	**Univariate Analysis**		**Multivariate Analysis**	
**Variable**	**β (95% CI)**	***p*-Value**	**β (95% CI)**	***p*-Value**
**FRS**				
M30	0.27	0.0008	−0.009	0.91
M65	0.31	0.00012	0.14	0.08
FLI	0.41	<0.0001	0.39	<0.0001
**SCORE2**				
M30	0.27	0.0006	0.01	0.86
M65	0.34	<0.0001	0.20	0.02
FLI	0.33	<0.0001	0.30	<0.0001
**FLI**				
M30	0.53	<0.0001	0.41	0.0001
M65	0.44	<0.0001	0.23	0.03
	**FLI ≥ 60 PREVEND Participants (n = 112)**			
	**Univariate Analysis**		**Multivariate Analysis**	
**Variable**	**β (95% CI)**	***p*-Value**	**β (95% CI)**	***p*-Value**
**FRS**				
M30	0.09	0.43	−0.09	0.54
M65	0.19	0.05	0.20	0.12
FLI	1.2	0.003	1.15	0.007
**SCORE2**				
M30	0.09	0.48	−0.20	0.20
M65	0.28	0.01	0.37	0.01
FLI	0.98	0.03	0.81	0.08
**FLI**				
M30	0.02	0.27	0.004	0.88
M65	0.03	0.12	0.032	0.28
	**FLI < 60 PREVEND Participants (n = 200)**			
	**Univariate Analysis**		**Multivariate Analysis**	
**Variable**	**β (95% CI)**	***p*-Value**	**β (95% CI)**	***p*-Value**
**FRS**				
M30	0.13	0.17	0.01	0.91
M65	0.14	0.19	0.09	0.41
FLI	0.37	<0.0001	0.36	<0.0001
**SCORE2**				
M30	0.20	0.04	0.10	0.43
M65	0.20	0.06	0.11	0.24
FLI	0.31	<0.0001	0.06	<0.0001
**FLI**				
M30	0.27	0.01	0.25	0.02
M65	0.14	0.23	0.04	0.70

β unstandardized regression coefficient; CI, confidence interval; FRS, Framingham risk score; SCORE: Systematic COronary Risk Evaluation; and FLI, fatty liver index. Because of the no normal distribution of all considered variables, data were transformed into logarithmic values.

## Data Availability

Data are available upon reasonable request.

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
