# Peer review of "Serum Level of Cytokeratin 18 (M65) as a Prognostic Marker of High Cardiovascular Disease Risk in Individuals with Non-Alcoholic Fatty Liver Disease"

_biomolecules, 2023, doi:10.3390/biom13071128_

Round 1

Reviewer 1 Report

Biomolecules-2482673

Serum Level of Cytokeratin 18 (M65) as a prognostic marker of high cardiovascular disease risk in Non-Alcoholic Fatty Liver Disease individuals 

Author: Sabrina Pagano1, Stephan J. L. Bakker3, Catherine Juillard2 and Robin P. F. Dullaart4, Nicolas Vuilleumier

The authors explored the discriminant accuracy of Cytokeratin 18 (CK18, including M65 and M30 forms) for an elevated fatty liver index (FLI) as a validated proxy of Non-Alcoholic Fatty Liver Disease (NAFLD), and cardiovascular disease (CVD) risk in the general population.

The authors evaluated both M65 and M30 in randomly selected samples from 312 participants of the PREVEND general population cohort. They used FLI ≥60 to indicate NAFLD. They used Framingham Risk Score (FRS) and the SCORE2 to estimate the 10-year risk of CVD. To analyze the data, the authors adopted Receiver Operating Characteristic (ROC) curve, linear/logistic regression models, and Spearman's correlations.

The authors found intricate associations between CK18, FLI, and CVD risk scores. They found that M30 was the only independent predictor of FLI≥60, M65 discriminated best against NAFLD individuals at very-high 10- years of CVD risk according to SCORE2 (AUC:0.71; p=0.001). They found that values above the predefined manufacturer cut-off (400 U/l), were associated with an independent 5-fold increased risk (adjusted odds ratio: 5.44, p=0.01), with a negative predictive value of 93%. 

The authors concluded that NAFLD is associated with an increased CVD risk and indicated that CK18 M65 is a candidate biomarker to identify NAFLD individuals at low CVD risk in the European general population.

Comments:

Major concerns:

1.     Is Cytokeratin 18 an appropriate candidate used in this study? The authors randomly selected 312 individuals with available fasting serum aliquots stored at -80ï‚°C. Whether Cytokeratin 18 keeps as it was in its initial state or differentially cleaved in a caspase-dependent manner or was simply degraded during storage is critically important. Some sort of verification that Cytokeratin 18 remains stable during storage may be required otherwise any conclusion would be in vain.

Minor:

1.      Title needs to be corrected to “Serum Level of Cytokeratin 18 (M65) as a prognostic marker of high cardiovascular disease risk in Non-Alcoholic Fatty Liver Disease individuals”

Moderate editing of the English language is required. 

Author Response

Biomolecules-2482673

Serum Level of Cytokeratin 18 (M65) as a prognostic marker of high cardiovascular disease risk in Non-Alcoholic Fatty Liver Disease individuals

Author: Sabrina Pagano1, Stephan J. L. Bakker3, Catherine Juillard2 and Robin P. F. Dullaart4, Nicolas Vuilleumier

The authors explored the discriminant accuracy of Cytokeratin 18 (CK18, including M65 and M30 forms) for an elevated fatty liver index (FLI) as a validated proxy of Non-Alcoholic Fatty Liver Disease (NAFLD), and cardiovascular disease (CVD) risk in the general population.

The authors evaluated both M65 and M30 in randomly selected samples from 312 participants of the PREVEND general population cohort. They used FLI ≥60 to indicate NAFLD. They used Framingham Risk Score (FRS) and the SCORE2 to estimate the 10-year risk of CVD. To analyze the data, the authors adopted Receiver Operating Characteristic (ROC) curve, linear/logistic regression models, and Spearman's correlations.

The authors found intricate associations between CK18, FLI, and CVD risk scores. They found that M30 was the only independent predictor of FLI≥60, M65 discriminated best against NAFLD individuals at very-high 10- years of CVD risk according to SCORE2 (AUC:0.71; p=0.001). They found that values above the predefined manufacturer cut-off (400 U/l), were associated with an independent 5-fold increased risk (adjusted odds ratio: 5.44, p=0.01), with a negative predictive value of 93%.

Major concerns:

1)    Is Cytokeratin 18 an appropriate candidate used in this study? The authors randomly selected 312 individuals with available fasting serum aliquots stored at -80°C. Whether Cytokeratin 18 keeps as it was in its initial state or differentially cleaved in a caspase-dependent manner or was simply degraded during storage is critically important. Some sort of verification that Cytokeratin 18 remains stable during storage may be required otherwise any conclusion would be in vain.

Response:

We appreciate the reviewer's comment. We aware the importance to address the stability of Cytokeratin 18, as a common and key pre-analytical factor applying per definition to any biomarker considered.

Previous published data have adequately addressed these pre-analytical issues for CK-18 (including caspase -cleaved form). In a nutshell, CK18 is stable up to six freeze-thaw cycles and is unaffected by long term storage at -80°C (doi:10.1158/1078-0432.CCR-07-0009; doi:10.1158/1078-0432.CCR-07-0009; DOI: 10.1007/s11095-005-9045-3). Manufacturer instructions even indicate that both form of CK18 are unaffected by long term storage at -20°C (https://www.tecomedical.com/en/Diagnostic-Assays-Reagents/Liver-Kidney-Apoptose/Liver/M30-Apoptosense-ELISA-PEVIVA/; https://www.tecomedical.com/en/Diagnostic-Assays-Reagents/Liver-Kidney-Apoptose/Liver/M65-ELISA-PEVIVA/ ).

Given the fact that the present study samples were stored for about 4 years at -80°C until analyses, thawed only once on melting ice, and that the median CK18 levels of our cohort was similar -and even rather higher- (M30:176.9U/L; IQR: 132.8-224.5 U/L; M65:173.1 U/L; IQR: 128.2-263.1 U/L) to those retrieved in two different studies including general and healthy blood donors populations (doi.org/10.3390/biom13040675; doi:10.1038/sj.bjc.6605175), we are convinced that all the pre-analytical requirements have been adequately respected and that no CK18 degradation occurred due to potential undetected pre-analytical protocol deviances. For these reasons, we consider that repeating preanalytical verifications to replicate available data in the literature will not further enhance the confidence we can have in the current results nor improve the quality of manuscript.

Minor:

  1. Title needs to be corrected to “Serum Level of Cytokeratin 18 (M65) as a prognostic marker of high cardiovascular disease risk in Non-Alcoholic Fatty Liver Disease individuals”

Response:

Done

Reviewer 2 Report

In this article, Serum Level of Cytokeratin 18 (M65) as prognostic marker of high cardiovascular disease risk in Non-Alcoholic Fatty Liver Disease individuals. The clinical data of serum CK18 as a biomarker for NAFLD and CVD patients is valuable in this field. However, there are some shortcomings and questions.

1)      There are a lot of serum biomarkers. In the introduction, other biomarkers should be introduced.

And why is CK18 unique?

2)      I read through the whole article. The results are shown only in the table. It is boring and unfriendly to the reader. It looks like to show some preliminary data.  Some results could be shown as graphs.

3)      In the discussion, the potential mechanism should be discussed.

Author Response

In this article, Serum Level of Cytokeratin 18 (M65) as prognostic marker of high cardiovascular disease risk in Non-Alcoholic Fatty Liver Disease individuals. The clinical data of serum CK18 as a biomarker for NAFLD and CVD patients is valuable in this field. However, there are some shortcomings and questions.

Response:

We appreciate the comment of the reviewer who states that measurement of CK18 may have value in the context of NAFLD and associated CVD risk burden

1)      There are a lot of serum biomarkers. In the introduction, other biomarkers should be introduced. And why is CK18 unique?

Response:

The interest of CK18 resides in the fact that it specifically reflects alterations in the cytoskeleton that occur during cell apoptosis, a process that takes place in steatotic liver, specifically during the development of NASH, and seems also to relate to atherogenesis and its complications in humans (PMID: 2472991, PMID: 19770641). These observations make CK18 a potentially important biomarker candidate to capture both NAFLD and atherogenesis-related complications which was the reason why this study was undertaken.

In the Introduction section of the revised manuscript, we have now indicated that there are other circulating biomarkers that may be useful to differentiate between NAFLD and NASH "page 2, line 67". CK18, measured using a commercially available ELISA (MK65: total CK18 and fragments; M30 fragments only) is an emerging biomarker that specifically reflects alterations in the cytoskeleton that occur during cell apoptosis, a process that takes place in a steatotic liver, in particular during the development of NASH. In addition, NAFLD and CVD are both manifestations of end-organ damage of the metabolic syndrome and a specific contribution of NAFLD to increased CVD risk is difficult to discern from the combination of these shared risk factors. CK18 could be also a reflection of apoptosis process that can happen in cardiomyocyte of NAFLD/NASH patients. A cardiomyocyte apoptosis process has already been described in the study from Narula et al. showing that heart failure can result from a variety of causes, including ischemic, hypertensive, toxic, and inflammatory heart disease and the cellular mechanisms responsible for the progressive deterioration of myocardial function observed in heart failure may result from apoptosis (Narula J et al. N Engl J Med. 1996 Oct 17;335(16):1182-9. doi: 10.1056/NEJM199610173351603. PMID: 8815940.

2)      I read through the whole article. The results are shown only in the table. It is boring and unfriendly to the reader. It looks like to show some preliminary data.  Some results could be shown as graphs.

      Response:

We appreciate the fact the reviewer took the time to go carefully in the details of the data presented in our manuscript. Linear, logistic regression and Spearman correlation analysis being very usual statistical approaches, and it is custom to represent such data in tabular form. To accomodate the critique of the reviewer we have now made two panels of graphs.

The first panels show CKM65 and CKM30 according to FLI, FRS and Score2 (New Figure 2). This figure is added to what is provided in the Tables. This Figure shows M30 and M65 levels according to dichotomized FLI, FRS and SCORE 2. Using non-parametric unpaired tests, CK(M30) and CK (M65) levels were higher in participants with an FLI ≥60 vs, an FLI<60 and in participants with a very high risk vs. participants with a low-moderate risk according to FRS, but were not significantly different upon dichotomization according to SCORE2.

The second panel shows Spearman correlation coefficients with liver function  tests and cardiometabolic biomarkers in the whole population studies as well as in those with and without an elevated FLI. This figure replaces the original Table 4 , and the exact P-values of these correlation coefficients are provided in a separate Supplemental Table (as supplemental Table S2).

As mentionned in the conclusion paragraph, line 417-420,we have stated that the current results are hypothesis-generating and need to be replicated at larger scale.

      3)      In the discussion, the potential mechanism should be discussed.

Response:

In the revised manuscript we have extended the discussion about the relevance and potential mechanisms whereby altered cytoskeleton biology as reflected by CK18 could in part delineate the intricate relationship between NAFLD/NASH and cardiometabolic disease. The corresponding text reads: NAFLD and CVD are both outcomes of end-organ damage caused by metabolic abnormalities commonly captured by metabolic syndrome components such as central obesity, high blood pressure, elevated glucose, elevated triglycerides and low HDL-c . It is challenging to determine the exact impact of NAFLD on increased CVD risk independently, as both conditions share common risk factors that contribute to this heightened risk [49]. The connections between NAFLD and CVD involve intricate and interrelated mechanisms that operate through multiple pathways simultaneously [49]. In this scenario CK18 (M30 and M65) could be a marker not only of the liver cell death but also of damaged cardiomyocytes that loss cellular integrity during response to abnormal cellular stresses such as endoplasmic reticulum stress and oxidative stress, which are recognized as key features of cardiometabolic disorders [26, 27]. Endothelial apoptosis can lead to endothelial dysfunction and the development of hypertension [50]. Apoptosis of cardiomyocytes is associated with both the aging process and chronic cardiac overload [51] and plays a substantial role in altering cardiac geometry and progressively deteriorating myocardial function, potentially leading to chronic cardiomyopathy and advanced heart failure [52]. A recent study reported that CK18 (M30) level was highly correlated to left ventricular (LV) diastolic dysfunction in adolescents with obesity [53] although CK18 (M65) had not been investigated. In addition, another study determined the relationship between the development of LV remodeling and CK18 (M30) but not CK18 (M65) in patients with anterior ST-segment elevation myocardial infarction [54]. In this study the cutoff used for CK18 (M30) was 144.9 U/L different from that we used (200 U/L) and they retrieved an AUC for CK18 (M30) level for predicting LV remodeling of 0.893 ) [54]. In an effort to assess patients with acute coronary syndrome (ACS), including unstable angina and acute myocardial infarction (AMI), as well as patients with stable angina, serum levels of CK18 (M30 and M65) were measured [23]. In addition, among patients with an acute coronary syndrome and stable angina, it was found that only CK (M30) levels accurately reflected the severity of coronary artery disease in acute myocardial infarction patients [23]. Collectively, these studies consistently indicate that M30 is a more reliable marker for myocardial acute events.

Our findings emphasize the association between CK18 (M65) and a higher risk of cardiovascular disease (CVD), likely due to M65 ability to detect cell death across various forms, including apoptosis, necrosis and autophagy and not only apoptosis as CK18 (M30) can detect.   

Reviewer 3 Report

This study measures the levels of cytokeratin 18, both forms M65 and M30, in individuals from the PREVEND general population of Groningen. The association between CK18, FLI, and CVD risk scores has been studied.

It needs some major revisions:

·      Revise all formats of the article, there are different types of font and different sizes of the font (authors, abstract section, tables, table feet, funding, acknowledgments, author contributions, conflicts of interest, references). It is an essential element that the paper follows the MDPI instructions. 

·      In the abstract section it needs a background sentence.

·      In line 46, add information about the progression to cirrhosis and hepatocarcinoma with specific references.

·      In line 69, change the format of “chronic kidney disease [20] to the correct one.

·      In the introduction section add more information about previous studies of cytokeratin 18 in NAFLD patients, such as: DOI: 10.3390/antiox9080759.

·      At the end of the introduction section it needs to specify the aim of the study.

·      In the methodology section, it needs to specify how venous blood samples were collected and how was isolated the serum samples.

·      In section 2.5, in which samples were determined all the biomarkers? Serum samples? Please, it needs to be added, it is elemental.

·      Discussion is too poor, it needs to be extended with more previous studies and compare them with your results.

·      Separate the limitations of the study in a different section, or put a title in bold, so that it stands out from the discussion.

·      A conclusion section is needed.

·      Reference section must be changed following the MDPI instructions.

Minors:

·      In line 69, change the format of “chronic kidney disease [20]” to the correct one.

·      Line 236: Correct “FI<60” to “FLI<60”.

Minor editing of English language required

Author Response

This study measures the levels of cytokeratin 18, both forms M65 and M30, in individuals from the PREVEND general population of Groningen. The association between CK18, FLI, and CVD risk scores has been studied.

It needs some major revisions:

  • Revise all formats of the article, there are different types of font and different sizes of the font (authors, abstract section, tables, table feet, funding, acknowledgments, author contributions, conflicts of interest, references). It is an essential element that the paper follows the MDPI instructions.

Response:

 We have scrutinized the lack of uniformity in font of the whole manuscript, and followed the MDPI instructions

  • In the abstract section it needs a background sentence.

Response:

   In the revised abstract we have now included a background sentence and the abstract has  now the format required from the journal.

  • In line 46, add information about the progression to cirrhosis and hepatocarcinoma with specific references.

   Response:

 The corresponding sentences have been amended and specific references have been included (Loomba R et al. Nat Rev Gastroenterol Hepatol. 2013 Nov;10(11):686-90. doi: 10.1038/nrgastro.2013.171; Benedict M, et al. World J Hepatol. 2017 Jun 8;9(16):715-732.     doi: 10.4254/wjh.v9.i16.715.)

  • In line 69, change the format of “chronic kidney disease [20] to the correct one.

Response:

 We apologize for the error in the format/font. Error corrected.

  • In the introduction section add more information about previous studies of cytokeratin 18 in NAFLD patients, such as: DOI: 10.3390/antiox9080759.

Response:

We agree with this critique of the reviewer. The paper of Monserrat-Mesquida M. et al. (Antioxidants (Basel). 2020 Aug 16;9(8):759. doi:10.3390/antiox9080759. PMID:  32824349;   PMCID: PMC7463614) as well as the systematic review by Sahebkar A. et  al. (J Cell Physiol. 2018 Feb;233(2):849-855) have been included now in the Introduction section.

         At the end of the introduction section it needs to specify the aim of the study.

 Response:

 The specific aims of the study have now been stated more clearly.

  • In the methodology section, it needs to specify how venous blood samples were collected and how was isolated the serum samples.

Response:

In the revised manuscript we have provided this information in much more detail, now reading: Participants were instructed to remain fasting from 22:00 PM the day before visiting the PREVEND facility. Blood was obtained by venipuncture from an antecubital vein after 15-30 min. rest. Blood samples were collected on melting ice and then allowed to clot. Serum was prepared by centrifugation at 1,000 x g for 10 min. Shortly thereafter, serum aliquots were stored at -80 0C. Samples were sent on dry ice to the laboratory of dr Pagano. Total storage time was about 4 years. Samples were thawed only on melting ice for measurement of CK18 and biochemical parameters required to calculated the FLI. For the purpose of the current study, we randomly selected 312 individuals with available serum samples for measurement of CK18 and biochemical parameters required to calculate the FLI and the CVD risk scores (page 3, line 106-115).

  • In section 2.5, in which samples were determined all the biomarkers? Serum samples? Please, it needs to be added, it is elemental.

Response:

 The other biomarkers were also measured in serum, except for glucose which was measured by dry chemistry at the PREVEND facility (Eastman Kodak). We added the sentence on page 4, line 153.

  • Discussion is too poor, it needs to be extended with more previous studies and compare them with your results.

Response:

 We further expand the discussion by presenting additional studies that explore the relationship between CK18 and cardiovascular disease (CVD). Please see our comment to reviewer 2. In the Discussion section of the revised manuscript we have added the following text: In the revised manuscript we have extended the discussion about the relevance and potential mechanisms whereby altered cytoskeleton biology as reflected by CK18 could in part delineate the intricate relationship between NAFLD/NASH and cardiometabolic disease. The corresponding text reads: NAFLD and CVD are both outcomes of end-organ damage caused by metabolic abnormalities commonly captured by metabolic syndrome components such as central obesity, high blood pressure, elevated glucose, elevated triglycerides and low HDL-c . It is challenging to determine the exact impact of NAFLD on increased CVD risk independently, as both conditions share common risk factors that contribute to this heightened risk [49]. The connections between NAFLD and CVD involve intricate and interrelated mechanisms that operate through multiple pathways simultaneously [49]. In this scenario CK18 (M30 and M65) could be a marker not only of the liver cell death but also of damaged cardiomyocytes that loss cellular integrity during response to abnormal cellular stresses such as endoplasmic reticulum stress and oxidative stress, which are recognized as key features of cardiometabolic disorders [26, 27]. Endothelial apoptosis can lead to endothelial dysfunction and the development of hypertension [50]. Apoptosis of cardiomyocytes is associated with both the aging process and chronic cardiac overload [51] and plays a substantial role in altering cardiac geometry and progressively deteriorating myocardial function, potentially leading to chronic cardiomyopathy and advanced heart failure [52]. A recent study reported that CK18 (M30) level was highly correlated to left ventricular (LV) diastolic dysfunction in adolescents with obesity [53] although CK18 (M65) had not been investigated. In addition, another study determined the relationship between the development of LV remodeling and CK18 (M30) but not CK18 (M65) in patients with anterior ST-segment elevation myocardial infarction [54]. In this study the cutoff used for CK18 (M30) was 144.9 U/L different from that we used (200 U/L) and they retrieved an AUC for CK18 (M30) level for predicting LV remodeling of 0.893 ) [54]. In an effort to assess patients with acute coronary syndrome (ACS), including unstable angina and acute myocardial infarction (AMI), as well as patients with stable angina, serum levels of CK18 (M30 and M65) were measured [23]. In addition, among patients with an acute coronary syndrome and stable angina,  it was found that only CK (M30) levels accurately reflected the severity of coronary artery disease in acute myocardial infarction patients [23]. Collectively, these studies consistently indicate that M30 is a more reliable marker for myocardial acute events. Our findings emphasize the association between CK18 (M65) and a higher risk of cardiovascular disease (CVD), likely due to M65 ability to detect cell death across various forms, including apoptosis, necrosis and autophagy and not only apoptosis as CK18 (M30) can detect.   

  • Separate the limitations of the study in a different section, or put a title in bold, so that it stands out from the discussion.

Response:

As requested, we put the limitations of the study as a separate paragraph with its own in bold title.

  • A conclusion section is needed.

Response:

 A separate conclusion paragraph has been highlighted more clearly in the revised Discussion section.

 Reference section must be changed following the MDPI instructions.

 Response:

Done

 Minors:

  • In line 69, change the format of “chronic kidney disease [20]” to the correct one.
  • Line 236: Correct “FI<60” to “FLI<60”.

Response:

 Done

Round 2

Reviewer 3 Report

Accept in present form